# Health by Words—A Content Analysis of Political Manifestos in the Portuguese Elections of 1975

**DOI:** 10.3390/ijerph18137119

**Published:** 2021-07-02

**Authors:** Paulo Mourao

**Affiliations:** Department of Economics & NIPE, Economics & Management School, University of Minho, 4700 Braga, Portugal; paulom@eeg.uminho.pt

**Keywords:** revolution, economic development, Portugal

## Abstract

The elections of 25 April 1975 had the highest participation to date in the Portuguese democratic experience. The competing forces organized electoral manifestos that covered a wide range of topics, with complex resources and analysis of the economic problems and the development of the Portuguese space. However, these manifestos’ contents are rich, there has not been a proper effort to study these political messages and reflect on the supply of answers provided by the competing parties to the collective needs assumed by the population, especially regarding Portuguese health. This work reviews these contents through a content analysis methodology, with the conclusion that the most common themes were related to the socioeconomic rights of Portuguese citizens and the need to boost development of different regions and fields, especially health. Other themes that the parties took on as central issues were those related to the war in the overseas territory and the stabilization of the emerging democratic process.

## 1. Introduction

What parties write is the current source of research in various scientific areas. The way the message is structured, the terms used, the expressions constructed, and the links between the themes are themselves relevant fields of analysis.

This fact underpins the focus of this analysis, which covers the 1975 Portuguese election, a crucial moment of popular response to the military coup of 25 April 1974 that changed the political regime in Portugal.

These elections, which took place on 25 April 1975, exactly one year after the Carnation Revolution, saw a record turnout of the electorate (with an abstention rate of around 8%). It was an electoral event that involved the participation of 12 party forces, with electoral manifestos based on a significant variety of issues. One of the most serious issues of these electoral manifestos was related to the health issue, a domain with critical values related to the needs of Portuguese people in this field. At the time, printed manifestos have been found to be a relevant channel for parties distributing their political messages, namely their suggestions of health policies.

This work will also analyze the content of the messages transmitted at that time by the official manifestos of these 12 parties. This content analysis will detail the attention of these parties to the dimensions of the economic and social development of the country.

The research question of this work is: were the messages conveyed by the competing parties at the Portuguese election of 1975 a strategical tool focused on the collective needs of Portuguese society? The central hypothesis of this work is, therefore, that the messages conveyed by the parties united the combined needs of society with the ideals of the various political forces.

In addition to testing the central hypothesis, in this work, we will also analyze the distribution of relevant terms along different socioeconomic dimensions and then proceed to an analysis of the similarity of the identified expressions.

As is easily seen, this is an innovative study that will show in detail the richness of the socioeconomic debate of this period of Portuguese democracy. The remainder of this paper is composed of three sections. Section 2 details the current state of the literature on electoral manifestos and characterizes the socioeconomic reality surrounding the 1975 election debate. Section 3 details the methodological effort that was designed to identify and analyze the frequency of terms and expressions in various manifestos, as well as to locate factors in the analysis and perform similarity analysis. Finally, Section 4 concludes the paper.

## 2. Manifestos and Political Manifestos—A Review of the Literature

### 2.1. Manifestos as an Object of Multidisciplinary Studies

Political manifestos are complex elements that have made possible a number of political, economic (Simone and Mourao [1]; Gentzkow et al. [2]; Grundler and Potrafke [3]), sociological (Milner and Judkins [4]), anthropological (Haupt [5]), and even psychological analyses. In practice, we can synthesize studies of the phenomenon into two main perspectives:(1)The manifesto as a “product of equilibrium” in the electoral market, composed of the ideological party offer, the main responsibility of the party, and the demand assumed by the electorate;(2)The manifesto as a social reflection of the surrounding context.

We will detail these areas of analysis.

#### 2.1.1. The Manifesto as Output in the Electoral Market

Just like how we conceive of a typical market, in an ideological and political market [6], we have a product of negotiation; we can easily include the components of political manifestos (ideas/political expressions) as the appropriate output. The structure of this argument is that the manifesto is a product “offered” by the different party groups as a constituency-based demand [7].

The establishment of manifestos as elements of equilibrium in the political market has its roots in such authors as Katz and Lazarsfeld [8] and Lasswell [9]. The fundamental idea comes from Katz and Lazarsfeld [8], who recognized communication as a specific response to a set of needs, whether to individual needs—as [9] so well referred to—or to group needs. Lasswell [9] developed his analysis along this line, reinforcing that no communication medium could be disconnected from the needs of the various transmitters and receivers, suggesting the five questions that have become famous in the analysis of communicative actions: Who—says What—To Whom—through What Medium—with What Intent.

In the specific case in question—that of the electoral manifestos in Portugal’s 1975 election—these political manifestos were important as elements of a compromise between a given ideological “offer” of the political parties and the “demand” of the electorate, which responded to the appeal with a record turnout (91.5% of voters voted) at that time [10,11,12,13].

As Dowey [14] or Belchior [15] have shown, electoral manifestos in such conditions are elements that result from the union of the needs of the electorate as well as the efforts of the insurgent forces of the emerging regime. For example, the manifestos observed in Fudge’s [16] analysis were revealed as being filled with expressions reflecting the socioeconomic needs of the electorate of a particular London borough.

#### 2.1.2. The Manifesto as a Social Reflection

McCombs [17] studied how media’s expressions contributed to defining the political agenda. In this sequence, Golding and Elliott [18] showed how the scheduling of political messages is the result of a meeting between the most latent social and political needs, the characteristics of societies, and the competition between emitters (be they political parties or economic groups).

Luria [19] also observed how political discourse is a transmissive process of information that uses a combined complex of means of communication that differentiates language in public functions. Electoral manifestos are the oldest elements in the diffusion of a political message, given the diffuse support of the masses and the focus on strong ideas that are usually compounded by the main concerns of society, the space of more eminent political debate, and the capacity of captivation of supporters/voters. In fact, it is not by chance that Luria [19] refers to the language of political manifestos as a monological, intentional language for a broad public.

Given the amplitude in the dissemination of manifestos, language itself becomes an element of its own analysis, as Budge et al. [20] explained. As Fraisse [21] states, “language is a behavior” itself. Thus, the choice of words, the sequence of messages, and the structure of themes, arguments, and counterarguments in political manifestos reflect the surrounding dimensions of the electoral market, including economic conditions [22], patterns of socioeconomic development [23], or even poverty levels [24].

Without the intensity of oral language, in manifestos primarily distributed by printed standards (pamphlets, newspapers, or partisans), the most frequent words are the most easily understood. Fraisse [21] refers to the presence of one or more general words, such as democracy, freedom, or constitutional rights. A strong correlation has been observed between the inductive words and the reactions induced, namely, the construction of a perception of possible answers to the problems of groups and individuals [2,3].

Following this general approach to electoral manifestos, we will focus on the case of the manifestos of the parties running in the Portuguese election of 1975. As we will lay out, the elections were the first ones involving the participation of an extended electorate, becoming one of the pillars of the current democracy in Portugal. Several works [25,26,27,28] have shown the dynamics of the political message broadcasted at the time in various channels, such as posters or graffiti [29], television, and radio space [30]. However, until now, no research has focused on the robustness of the electoral manifestos—typically, printed documents written by a few militants with the objective of synthesizing the basic ideas of the party movement for the election and capturing the attention of the electorate. These documents are unique elements, resulting from all of the dynamics operating during the electoral moment in question. Specifically, we refer to the pretensions to respond to the needs of the population, the guidelines of public policies, and the respect for the principles advocated by the Armed Forces Movement, the revolutionary organization that deposed the preceding dictatorial system.

### 2.2. The Socioeconomic Context of the Manifestos of the Portuguese Election of 1975

The 1975 election was considered in Portugal as the most important political moment after the Carnation Revolution (25 April 1974). Several authors [30,31] have interpreted it in a double dimension. In one perspective, it was a public recognition of the process of political democratization that began with the revolution. In another perspective, it was the first election that took place in Portugal with a universalized electorate [28,32,33].

The numbers that characterize this electoral process (which was initiated with the announcement of the electoral calendar by then-President of the Republic, Costa Gomes, on 10 February 1975) are still impressive: 2430 candidates from 12 parties ran for the 249 seats in the chamber that comprised the National Assembly (Assembly of the Republic), and they got involved in 21 days of campaigning, with thousands of rallies, briefings, and debates. On 25 April 1975, more than 6 million Portuguese citizens voted.

The 1975 elections were additionally a unique period for political analysis. Firstly, they were, in Portugal, the first elections held after the military coup of 25 April 1974. Therefore, they are a unique moment of perception of how the generality of Portuguese society in 1975 interpreted the various currents and countercurrents involve in the military coup of 25 April 1974. Second, they were the first elections after decades of a corporate dictatorship that significantly restricted electoral participation and political debate. They are, therefore, also an important point of analysis of how a society that grew up in the context of clandestine political debate saw itself in the possibility of directly intervening in the composition of the National Assembly and in the resulting choices for the government of the country. Finally, as will be highlighted below, these elections represented a broad space for debate on the country’s structural problems, but also on the solutions, namely in one of the most deficient fields: the field of health.

The analysis of this electoral moment has already produced debate among several authors [30,31]. The focus of these analyses varies. However, so far, no integrated debate has been held on the content of the electoral manifestos of the various political forces that presented themselves to the electorate at this historic moment for the Portuguese democracy. We intend to bridge this gap with the following sections.

The legal framework of the 1975 elections had two primary documents:DL (Decree Law) 621/74 (Articles 21 and 27 in particular, which dealt with the way in which citizen/party groups participated in the electoral process). This DL was composed of three depending documents: DL 621-A/74 (focused on general eligibility of citizens), DL 621-B/74 (focused on the restriction of eligibility related to citizens having had political connections with the former dictatorship), and DL 621-C/74 (focused on the number of eligible citizens for the parliamentary seats).DL 93-c/75 (which defined the electoral capacity and eligibility of citizens).

According to the legislation then in force (Decree-Law 621/74), a party was defined as a “[p]ermanent citizens’ organization constituted with the fundamental objective of democratically participating in the political life of the country and of competing in accordance with constitutional laws and its statutes and programs published, for the formation and expression of the political will of the people, intervening in the electoral process through the presentation or the sponsorship of candidacies.”

The election’s themes focused on a variety of scenarios [34,35]. These themes were explicit in parties’ manifestos. These manifestos tended to be written by each party’s directors considering the target public and the expectations of the supporters, combined with the overall needs of the society. The following issues were discussed: nationalized enterprises, preferential democratic systems, public health, women’s rights, free social assistance, struggle for the independence of the overseas territories, illiteracy, agrarian reform, opportunistic manipulation, control of production, (anti)communism, and the cost of living.

In Potrafke’s [36] work, the electoral manifestos are documents essentially related to the socioeconomic reality that involved the diversity of the voters. Let us detail the main points of the Portuguese economic development between 1970 and 1975.

The country’s economic and social structure in 1975 had its own characteristics. According to Costa [34], the majority of the population lived in an environment with rural characteristics, with only two cities having more than 500,000 inhabitants: Lisbon (where 700,000 inhabitants were concentrated) and Porto (with about 500,000 inhabitants). The domain of industrial production and income generated in industry was concentrated in Lisbon. Industry was then clearly the sector with the highest economic growth, with annual rates around 12% in areas such as electricity or water supply, in sharp contrast to the 0.5% annual growth rate of agriculture over the same period (1950–1973).

In addition to the demographic concentration around the country’s capital, Lisbon, and along the coastal strip, there was also a concentration of commercial societies and the best indicators of labor productivity in these territories [37,38]. A more updated work, [35] highlighted that between 1951 and 1988, Portugal failed to converge in labor productivity with European partners. From 1960 to 1985, Portugal’s accession to EFTA (European Free Trade Area) allowed it to develop an export sector based on the textile industry. Between 1961 and 1973, with Portugal’s accession to EFTA, Portuguese GDP (Gross Domestic Product) had grown at an average annual rate of 7%. Those were years when countries like Spain, Greece, and Portugal grew at a faster rate than the Irish rate. During this period, productivity per worker grew at an average rate of 6% per year in Portugal. This figure represented 1.9% above the European worker productivity growth rate. However, Portuguese productivity was low if we consider the value at the level—in 1961, the product per worker in Portugal represented only 28% of the European average; by 1973, the same figure had risen to 37% of the European average. However, as Pinho [35] mentions, the income per inhabitant in Portugal in 1974 was equivalent to 40% of the income per inhabitant in Europe in the same year.

Public spending in 1974 was very low, and, regarding health issues, Fernandes [37] identifies several severe public health needs: undernutrition in certain populations, low purchasing power to eating habits, and a low number of doctors able to work. In Portugal, each doctor had a potential target population of 1230 patients in 1970 (while, for example, the then-USSR had 460 inhabitants per doctor, and the United States had 670 inhabitants for each physician).

The birth rate was 19.6%, already showing a decreasing trend, while the mortality rate was 11%. In 1974, the infant mortality rate was one of the highest in Europe (40 per thousand). The proportion of the senior population (over 65) was 9.7%.

Fernandes [37] showed how the quality of housing was deficient, with the prevalence of low-quality housing due to overcrowding of the available space or due to problems of the structure of the buildings. [37,38] recognized that about a quarter of families lived in poor-quality conditions. Electricity consumption (kWh per capita) was 590 kWh in Portugal in 1970, when the USA was twelve times more [37].

### 2.3. Research Question and Current Hypothesis: Manifestos as Combined Products

As pointed out by Lancelot [39], Domenach [40], or Costa [41], the space of political debate in periods of transition between regimes can be analyzed as the meeting space of three types of messages:Messages of evident collective needs,Messages of potential collective needs,And messages of structuring ideals.

The political debate is commonly based on strategies of optimizing the potentiality of each actor’s message (and simultaneously based on minimizing the efficacy of the opponents’ messages). Therefore, the previous triangle of forces has also been observed in transitional periods of political party regimes [39,40]. The research question of this work is therefore: are the messages conveyed by the competing parties at the Portuguese election of 1975 a strategical tool (based on the political ideals of these competing parties), and were these messages trying to reply to the collective needs of Portuguese society (especially regarding the needs of health policy)?

Thus, following these arguments, we can test how the electoral manifestos of the parties competing in the 1975 election obeyed this structure. So, our hypothesis is that the messages conveyed by the parties united the combined needs of society with the ideals of the various political forces.

#### 2.3.1. Obvious Collective Needs

As evidenced by the work of Gaspar [42] and Fernandes [37], the emerging country of the Carnation Revolution had evident collective needs. The issues of school attendance and illiteracy were combined with health services being concentrated in large urban centers and only accessible to a minority of the population. In addition, infrastructure—both transport and communication—revealed obvious gaps. Furthermore, the nationalization processes coming from the MFA Program generated turbulence in the financial and banking markets, which led to a panic climate of depositors and investors [43].

#### 2.3.2. Potential Collective Needs

Within the group of potential collective needs were those rising from the intentions of granting independence to the overseas territories, which led to various doubts on the part of the Portuguese living in these spaces and their relatives (both in European territories and in these spaces). Several authors emphasized this climate of collective anxiety, which would materialize into human flows in subsequent years [29]. Therefore, the first potential collective need in 1975 was the independence of the overseas territories and the management of diplomatic, commercial, and demographic relations.

The second potential collective need was related to the direction following the concept of freedom that had guided the April Revolution. There was a great desire to materialize the liberty and rights enjoyed in other Western countries (such as freedom of expression and association), but there were also fears that the sense of freedom to be implemented might bring a repression of values that most of the Portuguese saw as their own. In particular, the issue of religious freedom and pressure on Catholic entities was viewed with increased attention [44].

#### 2.3.3. Structural Ideals

As mentioned by several authors [45], the self-titled “Motor of the Portuguese Revolution” was the “Movement of the Armed Forces” (MFA, from the Portuguese “Movimento das Forças Armadas”). This group directed the Revolution of 25 April 1974 and guided the political life of Portugal to the election of 1975 (and continued to be an impressive ideological force for many more years). Several studies have focused on the thinking of the MFA [46]. Based on the various texts signed by MFA leaders (MFA “Motor of the Portuguese Revolution”, 1975) and in complementary literature [47], we can synthesize the focus of the movement into the following points, converging with the so-called Revolutionary Program of the MFA. First, it sought to control inflationary tensions, reduce the cost of living, and increase the quality of life of the population. Second, it sought a political solution to the problem of wars overseas and launched an overseas policy that would lead to peace. Third, it intended to fight industrial and financial monopolies that were blamed for the imbalance in income distribution. Finally, the transverse lines of the various documents issued by the MFA in the months following 25 April 1974 underline the objective of developing the depleted country, solidifying democratic institutions through the famous three Ds: development, democratization, and decolonization.

## 3. Methodological Section—Empirical Analysis and Discussion

So far, a structured analysis of political manifestos of the 1975 election has not been done. Therefore, we want to do this research to fill that gap. A content analysis will provide a refreshed enlightenment on works like Carapinha et al. [48] or Gomes [29], which partially explored the richness of the messages of Portuguese parties written in political manifestos. In order to explore the quality of the content of the consulted party manifestos, we will follow the indications of authors like Robert and Bouillaguet [49]. Thus, we carried out a content analysis which, following Gavioli and Mourao [50], was structured in three main steps: analysis of the frequencies of the identified words, analysis of the association between these words, and, finally, analysis of the discourse structure.

As a first step, we analyzed the frequency of the identified relevant words. As a relevant word or relevant term, we identified those words in their nuclear form (that is, we focused on names and did not consider diminutives of names or focus on the infinitive of verbs, and we did not consider the different verbal forms of a given verb). We followed, in this perspective, Guessard [51]. As several authors have demonstrated [50,52,53], if the transcribed speech faithfully reflects the message reported by the sender, then the very frequent presence of a given word shows the importance of the associated problem for the issuer. Other works, such as Domenach [40], show that the sender, when using a word several times, expects the recipients of the message to share with him the importance of the problem and the search for solutions.

Table 1 and Figure 1 summarize the most frequent words in the party manifestos of 1975. There we found that terms like “social”, “right(s)”, or “economic” tended to be used very frequently by the various parties. We will comment on these words below.

### Empirical Analysis

The main source of our data was the compilations of Carapinha et al. [48], Ribeiro de Mello [54], and Abreu [55]. These elections saw the participation of 12 parties: MDP (Movimento Democrático Português), PCP (Partido Comunista Português), PPD (Partido Popular Democrático), PS (Partido Socialista), FEC (Frente Eleitoral de Comunistas), FSP (Frente Socialista Popular), LCI (Liga Comunista Internacionalista), MES (Movimento da Esquerda Socialista), CDS (Centro Democrático Social), PPM (Partido Popular Monárquico), PUP (Partido de Unidade Popular), and UDP (União Democrática Popular).

The 12 parties presented electoral manifestos that included content in the following thematic areas. We followed the 32 areas identified by Ribeiro de Mello [54] and Abreu [55]: Public Administration, Colonies, Social Communication, Concordat, Disarmament, Regional Development, Economy, Education, Elections, Family, Armed Forces, Strike, Housing, Scientific Research, Justice, Youth, Political Parties, Industrial Policy, International Politics, Monetary Policy, Agrarian Reform, Tax Reform, Wages, Health, Social Security, Trade Unions, Labor, Transport, Tourism and Urbanism, Single Liberties, Environment, and Monopolies. Comparing these areas with those discussed in Section 2.2 (The Socioeconomic Context of the Manifestos of the Portuguese Election of 1975), we can confirm that these areas detailed the major fields we discussed in Section 2.2 (where we provided an overall insight into the socioeconomic challenges of Portuguese society in 1975.

The majority of the parties responded to the various thematic areas, although to varying extents. The parties that gave answers in more detail were PS, PPM, PPD, and PCP.

In order to analyze the meaning and intensity of the political messages, we will begin by evaluating the frequency of the most quoted words. Based on the techniques derived from the free association of words, several fields of analysis show how an analysis of the most common terms to the answers (so-called commonalities) allows the extraction of several pertinent observations. The software chosen for this research was Iramuteq 0.7 Alpha 2 (developed by Pierre Ratinaud, LERASS, Toulouse, France).

The original documents, party manifestos, were written and consulted in Portuguese. They were then fully translated and proofread by a professional English editor.

The textual analysis process of the Iramuteq software has already been detailed in the work of Gilles et al. [56]. Generically, it has the following steps:
(1)Editing the document to be analyzed in a .txt file. The texts of each party are then separated by theme. The final document, including all manifestos from all parties with all topics covered, was recorded with a .txt extension.(2)This .txt file is then loaded into the Iramuteq software. The software then allows the selection of various parameters, namely the identification of the source language dictionary, the analysis of each word, or the reduction of each word to its root form (for example, the reduction of a verb form to its verb in the infinitive). The software also allows the identification of the priority word class (active). In our case, the active words were only three: verbs, nouns, and adjectives. This procedure minimizes, for example, problems arising from translation, while also minimizing the difference in frequency/quotation of the number of words in a sentence translated into different languages.(3)The software then allows the separate parameterization of several lines of analysis:
(a)Descriptive statistical analysis;(b)Content/words factor analysis;(c)Reinert method of distribution of messages by variables;(d)Analysis of similarity by variables and analysis of text centrality.


Each of these methods has specific steps that are covered in technical detail in Gilles et al. [56] or in Marpsat [57]. For example, the Reinert method (also known as the Alceste method) is discussed in Marpsat [57]. We can identify this method as being the basis of the analyses we did on political manifestos in 1975. In general, this method will assign to each observed word a pair of values of a distribution axis that works by proximity in the text. This step is a derivation of factor analysis. Thus, two words referred to several times in the same sentence or in close sentences will have similar values in terms of this distribution. The most common words or the most common themes will be on the central axis, and the less common words will be away from the central axis.

Jenkins [58] states that these observations demonstrate that the terms most common to a generality of individuals tend to be the objects most consistently focused on. In political discourse, this means that the most common terms in a message body show the focus of support groups on this political issue, given the contingency of the political moment. On the other hand, Noble [59] has shown how semantic/lexical diversity has important consequences in interpretation, from the diverse focus of terms to the heterogeneity of the issuer/receptor pair.

As can be seen in Table 1 and in Figure 1, the five most frequent terms are “social,” “right,” “economic,” “national,” and “political.”

The terms that were at the center of the political debate in the 1975 election were terms centered on developmental issues, as well as the extension of citizens’ social rights [60]. This agrees with our central hypothesis. Other terms with significant frequency prove this focus (worker, public, and company).

Representations of structural relations between groups of individuals/words in descending hierarchical classification figures are common. These figures are read from top to bottom (or from left to right for most dendograms). In our case, the corpus of the studied manifestos was firstly divided in two sub-corpuses (sub-corpus 1 with classes 2 and 3; sub-corpus 2 with classes 1 and 4). The process of inner division stops when it becomes non-significant at any other partition. Each class is composed of the individuals/terms that have a significant strength (association/coexistence in the sentences). A detailed explanation is provided by Gilles et al. [56]:


*“According to this method, the software first establishes a list of the vocabulary in the entire text, reducing words to their roots, on the basis of pre-established dictionaries (e.g., nurse and nursing reduced to a common entity nurs+). It then constructs different patterns of vocabulary distribution in order to identify discourse classes; these patterns are obtained by using automatic iterative descending hierarchical classifications to the analysed text. In other words, on the basis of their co-occurrences, pairs of words and sentences that are statistically frequently associated are gathered into the same class of discourse, and words that are less frequently associated form distinct classes. Chi-square tests provide a statistical indication of the strength of the association between vocabulary and classes: for a given class, words or excerpts that are statistically over-represented are referred to as typical, whereas those that are statistically under-represented (but relevant for other classes) are referred to as anti-typical. It is then up to the researcher to label the classes according to his or her interpretation of typical or anti-typical words or excerpts. By computing the χ2 test, the software estimated the strength of associations between classes of discourse and modalities of the following variables, extracted from the [sources].”*


We also ran additional figures, like the descending hierarchical classification. The dendrogram of descending hierarchical classification (Figure 2) enables us to understand each of the expressions and words present in the political manifestos, analyzing them from their locations and social insertions.

We verified that the corpus was divided into two main subgroups (class 2/3 versus class 1/4). The upper subgroup was divided into two. This division resulted in classes 2 and 3. The lower subgroup was divided in two others, which were class 1 and class 4.

Class 2 explained 26.7% of all political manifestos. It comprised terms like freedom (liberdade), Portuguese (Português), or territory (Território). This class was particularly related to the global positioning of the country in the world and its relations with other countries and peoples.

Class 3 explained 25% of the manifestos’ texts and was centered on terms linked to international relations, particularly the diplomatic efforts and cooperative policies regarding the African countries.

Class 4 explained the largest portion of the text, almost 30%. It focused on social policies, namely health policy, education policy, and related professional labor conditions. Class 1 was comprised of terms related to economic activities, especially the rural and agricultural activities that had a significant expression at the time [42].

These findings corroborate authors like Potrafke [36] and Costa [41], who claimed that the most common topics in political manifestos regard the population’s most important concerns at the time, which could secure votes for the party authoring the manifesto.

Then, in methodological terms, we established close relationships between dimensions of the texts under analysis. For this purpose, we will use factor analysis [61] and similitude analysis.

Factor analysis follows a data analysis technique that is already widely used. In this technique, the observations (in our case, the words collected in the party manifestos) are distributed according to score loadings [61,62]. These score loadings allow the distribution of the observations by latent dimensions, generally two factors. In our case, factor 1 associated the distribution of the words for international versus national issues (international issues on the left, national issues on the right), and factor 2 associated the distribution of the same words for individual (upper level) versus collective insights (lower level).

As Oliveira [63] states: “the closer the elements arranged in the plane, the more they speak of the same things.” As Oliveira [63] also notes, the presence of groups of terms/words in their own quadrants shows that each class encompasses specific semantic contexts. Moreover, the arrangement of groupings in opposite poles in the plane of the axes does not necessarily indicate a relation of semantic opposition of those same groupings.

The results of the factor analysis of the set of texts allowed the construction of Figure 3.

We found that four groups dominated in the responses (converging to Figure 2). The first was a core group with respect to the first factor, consisting of terms like “Freedom”, “Economy”, “Company”, “Production”, and “Control”, which are identified in green and red. This group included authors such as Freire [64], who showed how the electoral debate in 1975 revolved around the socioeconomic development needs of the country, something the majority of political actors considered as inextricably linked to the process of affirming the political freedoms associated with the Carnation Revolution.

Another prominent group was related to the set of content concerning the dimension of social development of the country. This group is marked in purple in Figure 3 and revolves around terms like “Education”, “Work”, “Salary”, “School”, and “Family”. Authors who focused on this topic in the political debate after the 25th of April include Santos [65] and Costa [41].

The fourth group was related to terms regarding concern with the dimension of the country’s international positioning. Here, we found terms like “Peace”, “People”, and “Relation” alongside “Portuguese” and “Independence”. This set—signaled in blue in Figure 3—reflected one of the major concerns that Portuguese society had at the time: the transition of powers in the overseas territories and the implications for the families of citizens of Portuguese nationality who lived in the territories during the process of independence. Given its distribution in Figure 3, we can interpret it as a very specific group within the two factors, along with the group of terms identified in purple.

Following Oliveira [63], the terms closest to the center of the graph represent those closest to the other thematic groups. According to Figure 3, practically no restricted set of terms occupies the axis of the figure, which demonstrates how the themes were exposed in electoral manifestos with very well-defined terminologies. This type of exhibition is not new in printed electoral manifestos, which differs, for example, from the type of exhibition in a live debate.

One of the capabilities of the Reinert method is also transposing the exercise performed in Figure 3 and highlighting the focus of each respondent, or, in this case, each manifesto. Thus, we performed this extension that enabled us to create Figure 4. Therefore, differences in the manifestos of the different political parties on health issues are suggested on Figure 4. Additional differences on other issues can also be highlighted if required.

In Figure 4, the 12 parties represented by the respective symbols/logos are visible as well as their electoral scores obtained in 1975. The distribution of these 12 parties along the axes of Figure 4 is associated with the ordinates estimated by the Reinert method. Thus, they represent the focus of each party in relation to the issues in the distribution of Figure 3. For example, the Popular Socialist Front party (which won 1% of the vote in 1975) focused its manifesto on international affairs more than, for example, the MES party, also with 1% of the vote (which, in turn, had its message more focused on economic issues and economic control). In Figure 4, the core of health-related messages is also highlighted (centroid of 1.32 and 0.77, both in Figure 3 and Figure 4).

Thus, Figure 4 shows, in a very convergent way with the readings derived from Downs’ Median Voter, how the manifestos are located in the thematic centers, avoiding focuses or disproportionate distributions of the subjects. We can thus state that the majority of the electoral forces in 1975 took on an equitable message in terms of the distribution of focuses across the various domains. However, the most voted for parties tended to present a message closer to health issues (or with a little additional emphasis) than the least voted parties. In fact, 64% of the electorate favored the two parties whose manifestos gave a closer prominence to health issues in 1975, which indicates the importance of health problems among the electorate in question.

In addition, the parties that presented themselves in the Portuguese legislative elections of 1975 signed the respective programs also focusing on specific health problems. Given the emphasis that we are placing on health, we refer to the fact that, in general, these competing parties focused on aspects such as the need to develop a public health service/system, the priority to improve the living conditions of the Portuguese people, the serious situation of assistance to more exposed citizens (the unborn, children, and the elderly), as well as the need to ensure greater longevity of the resident population. Although there was a common idea in the parties of the need to develop health policies that should be extended to the population, it was observed here that the center and center-right parties (PS, PSD, and CDS) focused mainly on promoting hospital conditions and a medical work context, while the other parties (identified with the left wing) placed the priority on free medical assistance, as well as the extension of the health service to all localities. A list of the saliences observed by each party, in Portuguese, will be made available if requested from the author.

To complete our methodological procedure, we carried out an analysis of similitude. This type of analysis allows the deepening of the association of words [63,64,65]. In the case of analysis of similitude, the methodology used associates central words with the remaining words of the analyzed text, associating them according to centrality values. Thus, the researcher realizes what the pivot words are and what the peripheral terms are.

Additionally, as [66,67,68] have shown, not only the frequency of terms, but also the direct link between the terms, is very important to verify, as the latter indicates an association that is considered significant in the context of the message. In contrast, very distant links (for example, the existence of many terms between two randomly chosen ones) shows that the potential for association (whether of causality or simultaneity) is considered much weaker [68].

As the example below shows (Figure 5), the 1975 electoral manifestos associated “economic” (one of the central terms) with other terms like “development”, “power”, and “order”. “Social” was closer to “condition”, “work”, and “right”.

Thus, we clearly found a more detailed conception that economic development was seen as strictly associated with the changing reality of the various sectors of the country, particularly the political life and the role of the public sector. This seems to confirm the thesis that the manifestos were a composite product between the preferences of the electorate and partisan ideas [69]. In contrast, the other dominant sphere in Figure 5—the social dimension—was more associated with messages conveying rights, education, health, and the improvement of working conditions. This was also anticipated by such authors as Costa [41] and Lopes et al. [70].

## 4. Conclusions

This was the first focused work dealing with the electoral manifestos of 1975 in Portugal through a content analysis. This analysis highlighted the parties’ differences across their manifestos regarding health issues. We also performed factorial analyses and similitude analyses, which allowed us to identify groups of words/terms that occurred together.

We concluded that the manifestos of the 12 parties that competed for these elections were intended to attract the electorate (an electorate involved in the revolutionary process after 25 April 1974) by presenting programmatic lines that responded to the population’s needs. These lines were concentrated in certain thematic domains: the social arena with a particular relevance to health issues, economic development, design of appropriate public policies, and establishment of a solution to tensions in the Portuguese-speaking African states.

In a synthesis of the discussion, we can mention that authors such as Ward et al. [69], Costa [41], and Lopes et al. [70] were validated by the present study. These authors referred to the strategic role of the electoral manifestos as products of responses to the needs observed in three vectors: evident collective needs, potential collective needs, and structuring ideology. It has been clearly evidenced that those manifestos paying more attention to health issues have been those signed by the victorious political forces of these elections.

### Further Developments

In terms of further research, this paper allows for four lines of study. The first line is related to extending this work to the subsequent Portuguese legislative election, in order to observe the dynamics of terms and frequencies, as well as changes in factorial analysis and similarity of expressions. The second line concerns the possibility of international comparative analysis, such as an analysis using the same methodological resources, but focused on other post-revolutionary electoral moments. The third line involves extending this analysis to the speeches (whether of victory or defeat) of the party leaders after the electoral moments, in order to examine the convergent/divergent themes between those moments of circumstantial emotionality and the complexity of the constructions associated with the manifestos. A fourth line emerges from the focused analysis of the health issue; therefore, it is here suggested as a relevant avenue the extension of this work considering the content analysis of the health issue across several parties’ manifestos and across European polls.

## Figures and Tables

**Figure 1 ijerph-18-07119-f001:**
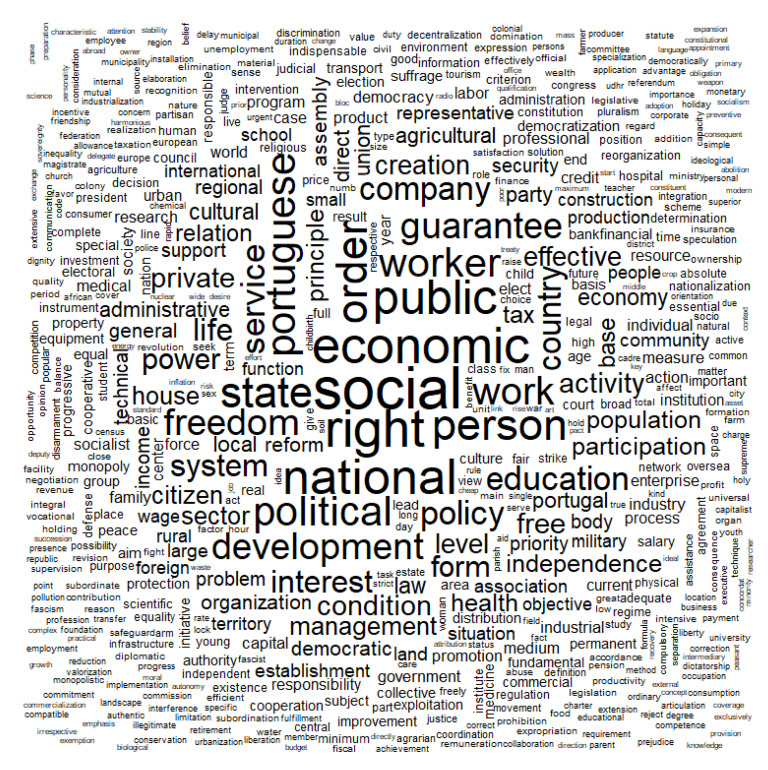
Word cloud of the collection of Portuguese parties’ manifestos (election, 1975).

**Figure 2 ijerph-18-07119-f002:**
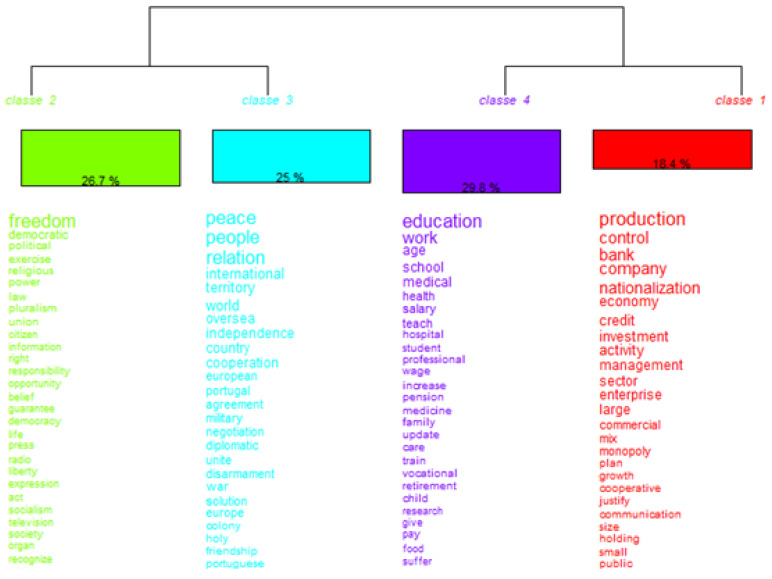
Descending hierarchical classification dendrogram.

**Figure 3 ijerph-18-07119-f003:**
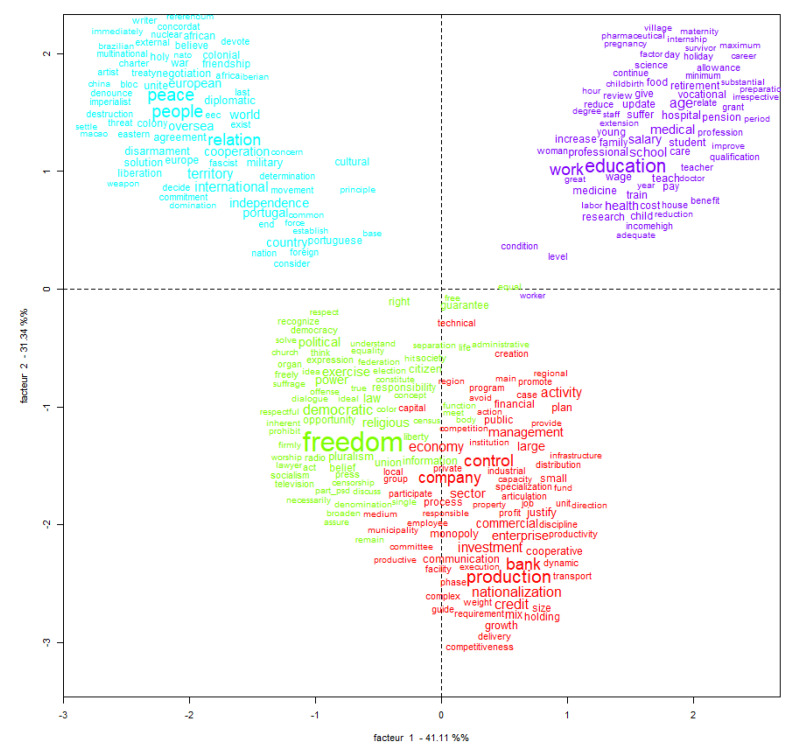
Factorial analysis of electoral manifestos (Portugal, 1975).

**Figure 4 ijerph-18-07119-f004:**
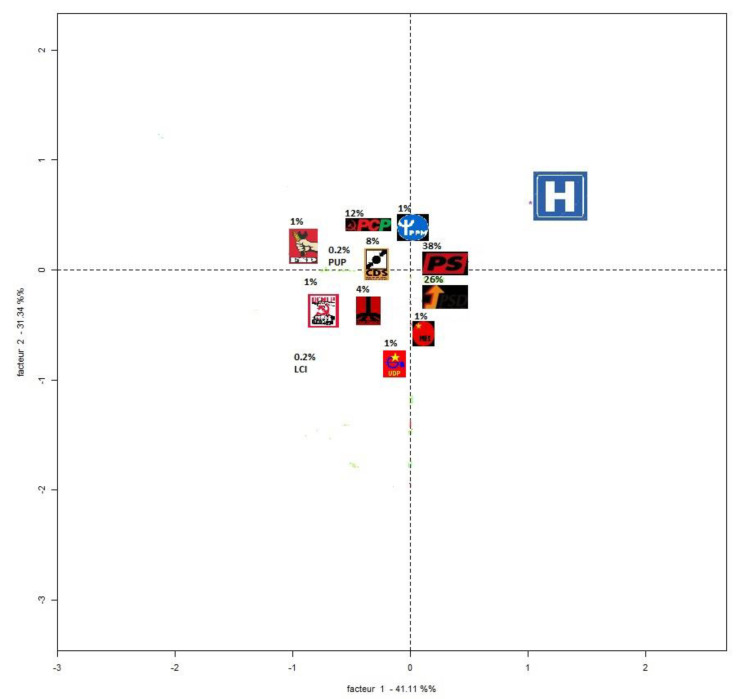
Factorial analysis of electoral manifestos (Portugal, 1975)—distribution for competing parties. Legend: MDP (Movimento Democrático Português), PCP (Partido Comunista Português), PSD (Partido Popular Democrático), PS (Partido Socialista), FEC (Frente Eleitoral de Comunistas), FSP (Frente Socialista Popular), LCI (Liga Comunista Internacionalista), MES (Movimento da Esquerda Socialista), CDS (Centro Democrático Social), PPM (Partido Popular Monárquico), PUP (Partido de Unidade Popular), and UDP (União Democrática Popular).

**Figure 5 ijerph-18-07119-f005:**
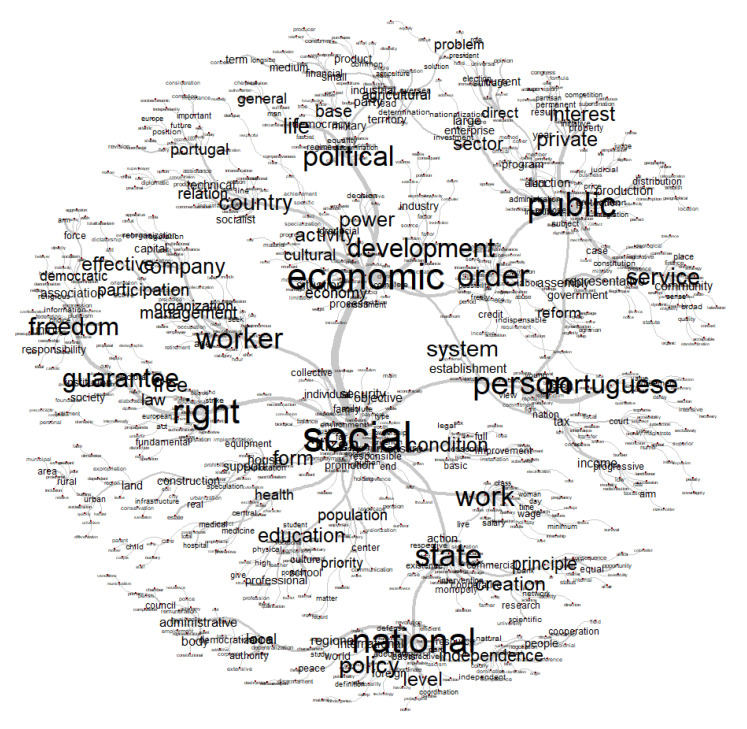
Analysis of similitude from the electoral manifestos (Portugal, 1975).

**Table 1 ijerph-18-07119-t001:** Mostly cited words in Portuguese manifestos (1975).

Word	#Cites
social	163
right	145
economic	129
national	126
politics	125
worker	115
to owe	114
public	113
form	111
company	103
medium	102
job	99
Portuguese	97
freedom	97
life	85
service	84
development	80
system	79
parents	72
creation	72
interest	70

## Data Availability

Data will be available under request to author.

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
