# Peer review of "Health by Words—A Content Analysis of Political Manifestos in the Portuguese Elections of 1975"

_ijerph, 2021, doi:10.3390/ijerph18137119_

Round 1
Reviewer 1 Report
IJERPH_1246626 Health by Words – A Content Analysis of Political Manifestos in the Portuguese Elections of 1975
The paper “Health by Words – A Content Analysis of Political Manifestos in the Portuguese Elections of 1975 “ uses text mining techniques to analyze the electoral manifestos compiled by the 12 political parties during the 1975 elections in Portugal. On the methodological side, the paper addresses an interesting and topical issue: how can we employ text analyses to get a better understanding about politics and historical events? I like the papers’ strategy to use text mining tool to examine what content the manifestos of the 1975 Portuguese election include. Regarding the more general contribution to the literature, however, I fear that the focus on manifestos of this particular election is quite narrow and definitely requires a better motivation. Why should we specifically focus on this particular election, and what can we learn more generally from the analysis? These points are touched only very briefly in the introduc-tory paragraphs and definitely require more discussion. I also have some additional comments on the paper. Below, I provide my specific comments.
MAJOR COMMENTS
(1) The contribution to the literature and the specific case of the 1975 Portuguese elec-tion: My main comment relates to the manuscripts’ contribution to the literature. I like the text mining analysis in general, but I fear that the focus on the particular election in Portugal in 1975 requires a much better motivation. This motivation/description should include the following: (1) why was the election so special? The authors describe the military coup in 1974 (Carnation Revolution), but the specific situation should be described much more precisely (potentially in a separate section; some of this discus-sion is “hidden” in section 2.2, but this is clearly the wrong place, because section 2 deals with a discussion of the literature, and this discussion should focus only on the methodological side), otherwise the non-Portuguese readership cannot follow your ar-guments. (2) Directly following from (1): the authors should use the arguments from (1) about why the 1975 election was special to better motivate their setting: What can we learn from this particular election (due to its special character) that cannot be learnt from other elections? This is a very important point. Finally, (3): The authors should then relate their findings to the existing literature and clearly describe their contribution to this literature. As we can learn from section 2 of the manuscript, there have been quite some papers that have analyzed political manifestos. The authors should hence clearly describe what can be learned from their paper that has not been shown elsewhere. What is the main takeaway for a general readership?
(2) Comments on the text mining analysis: The text analysis is conducted in English lan-guage. This is reasonable given that the standard software packages used in the au-thors’ analysis are trained on English dictionaries. But then I wonder how the authors
translated the manifestos (which are certainly written in Portuguese) into English lan-guage? This point requires further discussion, because I fear that the results are sensi-tive to the translation process. For instance, consider Table 1 that counts words. Does every English word refer to exactly one Portuguese word? I am not a Portuguese native speaker, but here is an example: The word “right”, the second-most used word in Table 1, means something like “just”. In Portuguese, this may be “justo/justa” or “equita-tivo/equitativa”. This may be a constructed example (again, I am not a native speaker) but I hope you see what I mean: There are synonyms and when we put together mul-tiple words, we would bias the word count, simply because we put several synonyms together to one word during the translation process. The same applies also for the remaining analysis, e.g. the word cloud in Figure 1.
In addition to my comment regarding language, the authors might want to describe some details on how they performed the text analysis. For example, which stop words did they exclude, and which commands did they use for their analysis (Figure 1 looks like the standard Python Code for text analysis, but this may be briefly described either in the appendix or in the Figure/Table notes).
(3) Sentiment analyses and differences across parties: Another very interesting point would be to consider differences across political parties. This can either be done by analyzing the words used by different parties or by sentiment analyses. I think the lat-ter would be a very interesting analysis. In particular, it would be interesting to see whether the manifestos of the winning/loosing parties differ. This could be related to the zeitgeist of Portugal in 1975 and the events around the military coup in 1974. I think that this additional analysis would provide the most interesting contribution to the literature.
MINOR COMMENTS
a) I suggest renaming Section 3. The title “Methodological Section” is not informative. Clearly describe in the title what you are doing in this section.
b) There are some grammatical errors in the paper, and I think the language can be im-proved at several places in the manuscript. Hence, I am convinced that the manuscript would benefit from careful English proofreading.
c) There are other papers that use similar text analyses tools like those employed by the authors (e.g. Gründler and Potrafke, 2020; see Gentzkow et al., 2019, for a survey). The authors might want to consider referring to these papers in their manuscript.
REFERENCES
Gentzkow, M., B. Kelly, and M. Taddy (2019): Text as data, Journal of Economic Literature 57.3: 535-74.
Gründler, K., and N. Potrafke (2020): Experts and Epidemics, CESifo Working Paper No.8556

Author Response
Now, there is an enlarged explanation about the translation process, about the content analysis considering the software Iramuteq, and about the Alceste Method.:” In order to analyze the meaning and intensity of the political messages, we will begin by evaluating the frequency of the most quoted words. Based on the techniques derived from the free association of words, several fields of analysis show how an analysis of the most common terms to the answers (so-called 'commonalities') allows the extraction of several pertinent observations. The software chosen for this research was Iramuteq 0.7 Alpha 2.
The original documents – party manifestos – were written and consulted in Portuguese (Carapinha et al (1974), Ribeiro de Mello (1975), and Abreu (1975)). They were then fully translated and proofread by a professional English editor.
The textual analysis process of the Iramuteq software has already been detailed in the work of Gilles et al (2017). Generically, it has the following steps:
1) Editing the document to be analyzed in a .txt file. The texts of each party are then separated by theme. The final document, including all manifestos from all parties with all topics covered, was recorded with a .txt extension.
2) This .txt file is then loaded into the Iramuteq software. The software then allows the selection of various parameters, namely the identification of the source language dictionary, the analysis of each word or the reduction of each word to its root form (for example, the reduction of a verb form to its verb in the infinitive ). The software also allows the identification of the priority word class (Active). In our case, the active words were only three: verbs, nouns and adjective. This procedure minimizes, for example, problems arising from translation, while also minimizing the difference in frequency/quotation of the number of words in a sentence translated into different languages.
3) The software then allows the separate parameterization of several lines of analysis:
- a) Descriptive statistical analysis;
b). Content/Words Factor Analysis;
c). Reinert method of distribution of messages by variables;
- d) Analysis of similarity by variables and Analysis of text centrality.
Each of these methods has specific steps that are covered in technical detail in Gilles et al (2017) or in Marpsat (2010). For example, the Reinert Method (also known as the Alceste Method) is discussed in Marpsat(2010). We can identify this Method as being at the basis of the analyzes we did on political manifestos in 1975. In general, this method will assign to each word observed a pair of values of a distribution axis that works by proximity in the text. This step is a Derivation of Factor Analysis. Thus, two words referred to several times in the same sentence or in close sentences will have similar values in terms of this distribution. The most common words or the most common themes will be on the central axis and the less common words will be away from the central axis.”
Following both of Reviewers’ suggestions, I also included an entire sub-section regarding differences across parties related to the focused domain – the Health domain. Please check paragraphs surrounding the Figure 4.
All minor issues have been fixed.
After this considerable revision, I recognize I am now submitting a much more appropriate version to be published by IJERPH.
Yours,
The Author.

Reviewer 2 Report
> First, I would like the author to show the evidence of a measure of
> heterogeneity/inequality in keywords across parties and time. This
> would be in important finding.
>
>
>
> Second, I would like to read explanations for differences in party
> health manifestos, how were they formed in Portugal, who are the yet
> actors influencing parties manifestos. Are certain parties closer to
> experts? What explains the differences? The role if vested interests?
>
>
> Third, the introduction needs to improve its motivation and the
conclusion should be strengthened. The role of political manifestos is key in competitive democracies, but we need to understand what their role is in Portugal.
Finally, more detail is need on the methods used.
Author Response
Dear Reviewer 2,
Thanks for your attention! As you can check, you provided relevant suggestions that I tried to follow in this version. As deserved, there is now an acknowledgement recognizing IJERPH reviewers’ contribution. Let me also express how stimulating I found IJERPH Editors’ and Reviewers’ expressions like “The paper “Health by Words – A Content Analysis of Political Manifestos in the Portuguese Elections of 1975 “ uses text mining techniques to analyze the electoral manifestos compiled by the 12 political parties during the 1975 elections in Portugal. On the methodological side, the paper addresses an interesting and topical issue: how can we employ text analyses to get a better understanding about politics and historical events? I like the papers’ strategy to use text mining tool to examine what content the manifestos of the 1975 Portuguese election include. (…)I like the text mining analysis in general (…)”.
Please check the following changes in this version.
Introduction and Conclusion have been updated, highlighting the motivation of the paper and the role of political manifestos.
Now, there is an enlarged explanation about the translation process, about the content analysis considering the software Iramuteq, and about the Alceste Method.:” In order to analyze the meaning and intensity of the political messages, we will begin by evaluating the frequency of the most quoted words. Based on the techniques derived from the free association of words, several fields of analysis show how an analysis of the most common terms to the answers (so-called 'commonalities') allows the extraction of several pertinent observations. The software chosen for this research was Iramuteq 0.7 Alpha 2.
The original documents – party manifestos – were written and consulted in Portuguese (Carapinha et al (1974), Ribeiro de Mello (1975), and Abreu (1975)). They were then fully translated and proofread by a professional English editor.
The textual analysis process of the Iramuteq software has already been detailed in the work of Gilles et al (2017). Generically, it has the following steps:
1) Editing the document to be analyzed in a .txt file. The texts of each party are then separated by theme. The final document, including all manifestos from all parties with all topics covered, was recorded with a .txt extension.
2) This .txt file is then loaded into the Iramuteq software. The software then allows the selection of various parameters, namely the identification of the source language dictionary, the analysis of each word or the reduction of each word to its root form (for example, the reduction of a verb form to its verb in the infinitive ). The software also allows the identification of the priority word class (Active). In our case, the active words were only three: verbs, nouns and adjective. This procedure minimizes, for example, problems arising from translation, while also minimizing the difference in frequency/quotation of the number of words in a sentence translated into different languages.
3) The software then allows the separate parameterization of several lines of analysis:
- a) Descriptive statistical analysis;
b). Content/Words Factor Analysis;
c). Reinert method of distribution of messages by variables;
- d) Analysis of similarity by variables and Analysis of text centrality.
Each of these methods has specific steps that are covered in technical detail in Gilles et al (2017) or in Marpsat (2010). For example, the Reinert Method (also known as the Alceste Method) is discussed in Marpsat(2010). We can identify this Method as being at the basis of the analyzes we did on political manifestos in 1975. In general, this method will assign to each word observed a pair of values of a distribution axis that works by proximity in the text. This step is a Derivation of Factor Analysis. Thus, two words referred to several times in the same sentence or in close sentences will have similar values in terms of this distribution. The most common words or the most common themes will be on the central axis and the less common words will be away from the central axis.”
Following both of Reviewers’ suggestions, I also included an entire sub-section regarding differences across parties related to the focused domain – the Health domain. Please check paragraphs surrounding the Figure 4.
Your suggestion regarding the relevance of a measure of heterogeneity in keywords along time has been inserted as a promising avenue for further research: In terms of further research, this paper allows for four lines of study. The first line is related to extending this work to the subsequent Portuguese legislative election, in order to observe the dynamics of terms and frequencies as well as changes in factorial analysis and similarity of expressions. The second line concerns the possibility of international comparative analysis, such as an analysis using the same methodological resources but focused on other post-revolutionary electoral moments. The third line involves extending this analysis to the speeches (whether of victory or defeat) of the party leaders after the electoral moments, in order to examine the convergent/divergent themes between those moments of circumstantial emotionality and the complexity of the constructions associated with the manifestos. A fourth line emerges from the focused analysis on the Health issue; therefore, it is here suggested as a relevant avenue the extension of this work considering the content analysis of the Health issue across several parties’ manifestos across European polls.”
In this version, I also detailed the relevance of this electoral moment: There is also an extension describing the manifestos’ authors: “The 1975 election was considered in Portugal as the most important political moment after the Carnation Revolution (April 25, 1974). Several authors (Palacios-Cerezales, 2003; Gomes, 2019) have interpreted it in a double dimension. In one perspective, it was a public recognition of the process of political democratization that began with the revolution. In another perspective, it was the first elections that took place in Portugal with a universalized electorate (Gaspar, 1983; Lisi, 2007).
The numbers that characterize this electoral process (which was initiated with the announcement of the electoral calendar by then-President of the Republic, Costa Gomes, on February 10, 1975) are still impressive. 2430 candidates from 12 parties ran for the 249 seats in the chamber that comprised the National Assembly (Assembly of the Republic), and they got involved in 21 days of campaigning with thousands of rallies, briefings, and debates. On April 25, 1975, more than 6 million Portuguese citizens voted.
The 1975 elections were additionally a unique period for political analysis. Firstly, they were, in Portugal, the first elections held after the military coup of April 25, 1974. Therefore, they are a unique moment of perception of how the generality of Portuguese society in 1975 interpreted the various currents and counter-currents involve in the military coup of April 25, 1974. Second, they were the first elections after decades of a corporate dictatorship that significantly restricted electoral participation and political debate. They are, therefore, also an important point of analysis of how a society that grew up in a context of clandestine political debate saw itself in the possibility of directly intervening in the composition of the National Assembly and in the resulting choices for the government of the country. Finally, as will be highlighted below, these elections represented a broad space for debate on the country's structural problems but also on the solutions, namely in one of the most deficient fields – the field of Health.
The analysis of this electoral moment has already produced debate among several authors (Palacios-Cerezales, 2003; Gomes, 2019). The focus of these analyses varies. However, so far, no integrated debate has been held on the content of the electoral manifestos of the various political forces that presented themselves to the electorate at this historic moment for the Portuguese democracy. We intend to bridge this gap with the following sections.
The legal framework of the 1975 Elections had two primary documents:
- DL 621/74 (Articles 21 and 27 in particular, which dealt with the way in which citizen/party groups participated in the electoral process). This DL was composed by three depending documents: DL 621-A/74 (focused on general eligibility of citizens), DL 621-B/74 (focused on the restriction of eligibility related to citizens having had political connections with the former Dictatorship) and DL 621-C/74 (focused on the number of eligible citizens for the parliamentary seats).
- DL 93-c/75 (which defined the electoral capacity and eligibility of citizens)
According to the legislation then in force, a party was defined as a "[p]ermanent citizens' organization constituted with the fundamental objective of democratically participating in the political life of the country and of competing in accordance with constitutional laws and its statutes and programs published, for the formation and expression of the political will of the people, intervening in the electoral process through the presentation or the sponsorship of candidacies."
The election’s themes focused on a variety of scenarios (Teodoro, 2007). These themes were explicit in parties’ manifestos. These manifestos tended to be written by each party’s directors considering the target-public and the expectations of the supporters combined with the overall needs of the society. The following issues were discussed: nationalized enterprises, preferential democratic systems, public health, women's rights, free social assistance, struggle for the independence of the overseas territories, illiteracy, agrarian reform, opportunistic manipulation, control of production, (anti) communism, and cost of living.
In Potrafke’s (2009) work, the electoral manifestos are documents essentially related to the socio-economic reality that involved the diversity of the voters. Let us detail the main points of the Portuguese economic development between 1970 and 1975.”
All minor issues have been fixed.
After this considerable revision, I recognize I am now submitting a much more appropriate version to be published by IJERPH.
Yours,
The Author.

Round 2
Reviewer 1 Report
Dear authors,
I have now thoroughly reviewed the revised version of your manuscript. I appreciate the effort you put into revising the manuscript and I think that the paper has improved a lot now. I have no additional comments or suggestions regarding the analysis and I think that the paper now makes an interesting contribution to the literature.
I spotted some typos and grammatical issues in the text (e.g. "Health policies" on Page 1 and "Regarding" on Page 5, which should all not be in capital letters). I strongly suggest going carefully over the manuscript and correcting these (and other) grammatical issues.
Author Response
Thanks for your Acceptance decision. Additional Minor issues have been fixed.
Reviewer 2 Report
I think he paper can be accepted now
Author Response

(The authors gave the same response as above.)
